# Tumor Infiltration with CD20^+^CD73^+^ B Cells Correlates with Better Outcome in Colorectal Cancer

**DOI:** 10.3390/ijms23095163

**Published:** 2022-05-05

**Authors:** Frederik J. Hansen, Zhiyuan Wu, Paul David, Anke Mittelstädt, Anne Jacobsen, Malgorzata J. Podolska, Kenia Ubieta, Maximilian Brunner, Dina Kouhestani, Izabela Swierzy, Lotta Roßdeutsch, Bettina Klösch, Isabella Kutschick, Susanne Merkel, Axel Denz, Klaus Weber, Carol Geppert, Robert Grützmann, Alan Bénard, Georg F. Weber

**Affiliations:** 1Department of General and Visceral Surgery, Friedrich-Alexander-University, 91054 Erlangen, Germany; Frederik.Hansen@fau.de (F.J.H.); Zhiyuan.Wu@fau.de (Z.W.); Paul.David@uk-erlangen.de (P.D.); Anke.Mittelstaedt@uk-erlangen.de (A.M.); Anne.Jacobsen@uk-erlangen.de (A.J.); Malgorzata.Podolska@uk-erlangen.de (M.J.P.); Kenia.Ubieta@uk-erlangen.de (K.U.); Maximilian.Brunner@uk-erlangen.de (M.B.); Dina.Kouhestani@uk-erlangen.de (D.K.); Izabela.Swierzy@uk-erlangen.de (I.S.); Lotta.Rossdeutsch@uk-erlangen.de (L.R.); Bettina.Kloesch@uk-erlangen.de (B.K.); Isabella.Kutschick@uk-erlangen.de (I.K.); Susanne.Merkel@uk-erlangen.de (S.M.); Axel.Denz@uk-erlangen.de (A.D.); Klaus.Weber@uk-erlangen.de (K.W.); Robert.Gruetzmann@uk-erlangen.de (R.G.); Alan.Benard@uk-erlangen.de (A.B.); 2Department of Pathology, Friedrich-Alexander-University, 91054 Erlangen, Germany; Carol.Geppert@uk-erlangen.de

**Keywords:** CD20, B-cells, CD73, colorectal cancer, neoadjuvant treatment

## Abstract

Immunotherapy has become increasingly important in the treatment of colorectal cancer (CRC). Currently, CD73, also known as ecto-5′-nucleotidase (NT5E), has gained considerable interest as a potential therapeutic target. CD73 is one of the key enzymes catalyzing the conversion of extracellular ATP into adenosine, which in turn exerts potent immune suppressive effects. However, the role of CD73 expression on various cell types within the CRC tumor microenvironment remains unresolved. The expression of CD73 on various cell types has been described recently, but the role of CD73 on B-cells in CRC remains unclear. Therefore, we analyzed CD73 on B-cells, especially on tumor-infiltrating B-cells, in paired tumor and adjacent normal tissue samples from 62 eligible CRC patients. The highest expression of CD73 on tumor-infiltrating B-cells was identified on class-switched memory B-cells, followed by naive B-cells, whereas no CD73 expression was observed on plasmablasts. Clinicopathological correlation analysis revealed that higher CD73^+^ B-cells infiltration in the CRC tumors was associated with better overall survival. Moreover, metastasized patients showed a significantly decreased number of tumor-infiltrating CD73^+^ B-cells. Finally, neoadjuvant therapy correlated with reduced CD73^+^ B-cell numbers and CD73 expression on B-cells in the CRC tumors. As promising new immune therapies are being developed, the role of CD73^+^ B-cells and their subsets in the development of colorectal cancer should be further explored to find new therapeutic options.

## 1. Introduction

Colorectal cancer (CRC) is the third most commonly diagnosed cancer and is the second leading cause of cancer-related death worldwide [1]. Standard treatments for CRC include radical resection with regional lymph node dissection by total mesorectal excision for rectal cancer or complete mesocolic excision for colon cancer. This may be complemented by (neo)adjuvant radio(chemo)therapy for locally advanced rectal cancer and/or systemic chemotherapy with adjuvant or palliative intent [2]. Over the past years immunotherapy has been under development in the treatment of CRCs, but only a limited group of patients benefits [3]. Therefore, discovering the functions and underlying mechanisms of novel immune checkpoint targets is of fundamental importance. Immune checkpoints are ligand-receptor pairs that exert stimulatory or inhibitory effects on immune responses [4]. Among the immune checkpoints, CD73 has gained considerable interest as a potential therapeutic target within the last decade. CD73, also known as ecto-5′-nucleotidase (NT5E), is one of the key enzymes catalyzing the conversion of adenosine monophosphate (AMP) into the immunosuppressive adenosine, which subsequently activates type 1 purinergic receptors to exert anti-inflammatory effects by inhibiting immune cell activity [5,6]. Malignancies such as CRCs exhibit increased levels of adenosine in the tumor microenvironment (TME), which suppresses the activity of cytotoxic T lymphocytes and natural killer cells, leading to tumor immune escape [7]. CD73 has been shown to be increased in tumor tissues in comparison to normal control samples, and higher CD73 is generally correlated to a worse prognosis in multiple types of human cancers, including CRCs [8,9,10,11,12]. However, a previous study revealed that a higher CD73 expression in tumor cells and stromal components of rectal adenocarcinoma correlated with a poor and favorable prognosis, respectively [12]. Hence, the exact effect of CD73 expression in various cellular compositions of the tumor microenvironment on patient outcomes is yet to be discovered. B lymphocytes, an important cellular component of the adaptive immune system, have been identified in CRC tumors and exert a significant impact on tumor progression and prognosis [13]. CD73 is expressed on human peripheral B-cells, but the role of CD73 on tumor-infiltrating B-cells, and its impact on clinicopathological characteristics of CRC patients, remains elusive [14,15]. Therefore, the aim of the present study was to analyze the expression of CD73 on CRC tumor-infiltrating B-cells and a variety of B-cell subsets, as well as its correlation to the available clinicopathological characteristics of the patients, to explore their possible clinical implications.

## 2. Results

### 2.1. CD73 on Different Cell Types in CRC

First, we performed H&E stainings on histology sections of native and tumor tissue samples to confirm malignancy (Figure 1A). In order to assess the general expression pattern of CD73 in the tissue samples, IHC staining of CD73 was performed. As shown by a group of exemplary graphs (Figure 1B), CD73 showed strong enhancement in tumor tissues compared to normal control tissues, and the positive CD73 staining was mostly aggregated in the tumor nests. Next, we investigated the CD45^+^ leucocyte number (normalized as cell number per gram tissue) by using multicolor flow cytometry analysis and found a significant increase in leukocytes in the tumor tissue (Figure 1C). Due to the increase in the cell count of CD45^+^ leucocytes and the enhancement of CD73 expression in the IHC staining of tumor tissues, we were interested in the CD73 expression on CD45^+^ leucocytes. By using multicolor flow-cytometry analysis we found that the CD45^+^CD73^+^ cell number was significantly increased, whereas the expression of CD73 on CD45^+^ leucocytes was decreased in CRC tumor tissues in comparison to the paired adjacent normal tissues (Figure 1D). As B-cells are our cells of interests, we first analyzed the number of CD20^+^ B-cells and found a significant increase in tumor tissues (Figure 1E). Next, we were interested in the CD73 expression on these tumor-infiltrating B-cells. We also found a significant increase in the cell count of CD20^+^CD73^+^ B-cells and a decreased expression in CD73 on CD20^+^ B-cells in tumor tissues (Figure 1F). Finally, we correlated the number of CD20^+^CD73^+^ cells and the MFI of CD73 on B-cells with tumor localization (Figure 1G). The CD20^+^CD73^+^ B-cell number was significantly elevated in tumors of the rectum, while in the right and left colon tumors only an increasing tendency was observed (*p* = 0.5850 and *p* = 0.2281, respectively). Moreover, the CD73 MFI on B-cells was significantly reduced in tumors of both the right colon and the rectum, whereas in the left colon tumors only a decreasing tendency was found (*p* = 0.2161). As a result of the observation of a consistent tendency for each group, we pooled all samples in our further analyses.

### 2.2. CD73 on Different B-Cell Subtypes

Subsequently, we looked into the expression of CD73 on several B-cell subtypes in a smaller prospective cohort. The gating strategy for identifying naive B-cells (CD20^+^IgD^+^CD27^−^), class-switched memory B-cells (CD20^+^IgD^−^CD27^+^), and plasmablasts (CD20^+^IgD^−^CD27^+^CD38^+^), as well as CD73 expression on different B-cell subsets, are demonstrated in Figure 2A. We found that the majority of CD73-expressing B-cells was allocated to class-switched memory B-cells in both tumor and blood samples, followed by naive B-cells; nevertheless, it seemed that plasmablasts expressed no CD73 at all (Figure 2A,B). Interestingly, we demonstrated that CD73^+^ B-cells expressed significantly lower IgM but higher IgG compared to CD73^−^ B-cells in the tumor samples (Figure 2C).

### 2.3. CD73 on B-Cells and Clinicopathological Characteristics

First, we divided the patients according to the median number of tumor infiltrating CD20^+^CD73^+^ B-cells (51,589 cells/g tissue) and found that patients with high infiltration had a significantly favorable overall survival rate compared to patients with low infiltration (Figure 3A and Table 1). 

Subsequently, we grouped the patients according to the status of lymph node involvement and did not find any difference between patients with involved (pN+, *n* = 17) and spared lymph nodes (pN0, *n* = 45) with respect to CD20^+^CD73^+^ B-cell number (Figure 3B). Concerning the distant metastasis of the tumors, we found significantly higher CD20^+^CD73^+^ B-cell numbers in localized (M0, *n* = 45) compared to metastasized (M+, *n* = 17) tumors (Figure 3C). To support our findings, that high infiltration of CD20^+^CD73^+^ B-cells is correlated with better survival and reduced risk of developing metastasis, we also compared the survival of patients with localized and metastasized tumors (Figure 3D). As expected, we found that patients with localized tumors had a significantly better survival rate. Next, we analyzed the CD20^+^CD73^+^ B-cell number among G1 (*n* = 3), G2 (*n* = 25) and G3 (*n* = 26) tumors and found no difference (Figure 3E). Finally, we demonstrated that both the CD20^+^CD73^+^ B-cell number and CD73 MFI on B-cells were significantly reduced in patients that received neoadjuvant therapy (*n* = 20) compared to neoadjuvant treatment-naive patients (*n* = 42) (Figure 3F and Table 2).

## 3. Discussion

Previous studies have established that the expression of CD73 in CRC tumors is higher than that in normal colorectal tissues [11,12], as confirmed by our analysis using IHC staining of CD73 and high-precision Fluorescence-activated Cell Sorting (FACS) showing increased CD45^+^CD73^+^ cell numbers. However, the MFI of CD73 on CD45^+^ leucocytes was decreased in CRC tumors. As previous studies have demonstrated the immunosuppressive and pro-tumorigenic role of CD73 in the TME of multiple types of cancers [16,17,18,19], higher CD73 on tumor cells indicates a tendency to escape immune surveillance.

B-cells have long been overlooked by most investigators in the study of anti-tumor immunotherapy until recently. B-cells comprise various subpopulations with highly complex developmental processes [20]. In general, mature B-cells recognize foreign antigens by their BCR complex, which in turn stimulates antigen-specific antibody responses and promotes B-cell differentiation into plasma cells and memory B-cells [21]. Apart from their role in antibody production, B-cells also serve as antigen-presenting cells and are required for T-cell activation and cellular immunity [22]. Tumor-infiltrating B-cells have been identified in various kinds of human malignancies [21]. Importantly, a recent study shows that multiple B-cell subtypes, especially activated and terminally differentiated B-cells, are detected in CRC tumors; while metastatic tumors are generally characterized by a lack of B-cell infiltration, the inhibitory regulatory B-cells could only be detected in advanced cancers and metastases [23]. Moreover, recent studies show that the presence of CD20^+^ B-cells is correlated with favorable pathology and improved survival in patients with CRC [13,24]. Therefore, we also correlated the cell number of CD20^+^ B-cells in tumor tissues with the clinical parameters as shown above, and found no significant differences at all. This might show the special role of CD73 expressing B-cells. Nevertheless, it is still controversial concerning the influence of B-cells on anti-tumor immunity, as preclinical studies show both pro-tumorigenic and anti-tumorigenic effects in different mouse tumor models [25]. Thus, further investigations into the effects of distinct B-cell subpopulations in the CRC TME are needed. 

CD73 expression on B-cells is believed to influence the immune response through multiple mechanisms. For instance, it is reported that by using in vitro cultures, activated B-cells suppress T-cell activity by down-regulating CD73, which subsequently inhibits adenosine production and results in AMP accumulation [14]. Another study, using a mouse colitis model, shows that B-cells inhibit inflammation by expressing CD73 and adenosine [26]. Apart from promoting T-cell effector response, the study by Forte et al. shows, with a mouse melanoma model, that administration of CD73 inhibitor also increases tumor-infiltrating B-cells, thus suppressing tumor growth [27]. The exact function and underlying mechanisms of down-regulated B-cell-specific CD73 expression in CRC tumors warrants further investigation.

Previous studies have revealed that CD73 is expressed on many B-cell subtypes in humans [28]. However, the expression pattern of CD73 on different B-cell subpopulations and the resulting functions in CRCs or any other solid malignancies are still elusive. By using multi-color FACS, we identified several common B-cell subtypes in CRC tumors and analyzed their CD73 expression pattern. CD73 was expressed on naive and class-switched memory B-cells, but not on plasmablasts in both CRC tumors and the blood of the patients, which is consistent with the study of Conter et al. [29]. We also found that CD73 expression was higher in class-switched memory B-cells compared to naive B-cells, and CD73^+^ B-cells expressed less IgM but more IgG compared to CD73^−^ ones, both indicating a positive correlation between B-cell class switch recombination and CD73 expression. Nevertheless, whether CD73 up-regulation is a cause or consequence of B-cell class switch recombination is still unknown. 

It has been reported that CD73 expression on different cell types within the TME exerts distinct influences on patient outcomes. A previous study shows that high CD73 on malignant epithelial cells of rectal adenocarcinomas correlates with a poor prognosis, whereas high CD73 in stromal components correlates with favorable pathological characteristics and overall survival [12]. In the present study, we were the first to demonstrate that higher CD20^+^CD73^+^ B-cell infiltration is correlated with favorable overall survival rates of CRC patients. As B-cells have long been overlooked by most investigators in the study of tumor immunology until recently, and the influence of B-cells on anti-tumor immunity is still controversial [25], further investigations into the underlying mechanisms that CD73^+^ B-cell infiltration favors patient survival are necessary. In addition, we found that fewer CD73^+^ B-cells infiltrated tumors which had metastasized. Previous studies report that CD73 deficiency is protective against pulmonary metastasis development of melanoma and prostate cancer cells following tail vein injection [17,18]. Contrarily, the CD73-adenosine axis is also shown to exert barrier-protecting functions in vascular endothelium and intestinal epithelium [30,31]. A recent study demonstrates that CD73-derived adenosine protects epithelial integrity by adenosine receptor-mediated actin polymerization [32]. CD73 is down-regulated in advanced-stage high-grade endometrial carcinoma, leading to disrupted endometrial endothelial barrier function, which favors the invasion and migration of the cancer cells [32]. The exact mechanism of how CD73^+^ B-cell infiltration is correlated to decreased probability of CRC tumor metastasis, however, needs further investigation. 

Our findings suggest that CD73^+^ B-cell infiltration might be used as a potential prognostic biomarker for overall survival and metastasis, and that patients at risk might require more aggressive treatment interventions. The observed reduction in both CD73^+^ B-cell number and CD73 expression on B-cells in CRC tumors that underwent neoadjuvant therapy suggests that neoadjuvant therapy inhibited CD73 expression on B-cells. As demonstrated by Saze et al., the down-regulation of CD73 on activated B-cells reduces their adenosine production, thus alleviating its inhibitory effect on B-cell functions [14]. A recent study shows that preoperative radiotherapy is associated with reduced infiltration of CD20^+^ B-cells [24], which seems to be consistent with our findings of reduced CD73^+^ B-cells. However, distinct from our observation of the correlation between neoadjuvant therapy and the reduction in B-cell-specific CD73 expression, some previous studies reported increased CD73 on tumor cell lines both in vitro and in transplanted tumor models following chemotherapy or radiotherapy [33,34]. It seems that neoadjuvant therapy changes both immune cell infiltration and CD73 expression on various cell populations in the TME, which needs further investigation. 

Limitations must be taken into consideration when viewing the findings of the present study. Due to the relatively low number of patients, we refrained from separate analyses of the different localizations. It is well known that the localization of the tumor has a significant impact on the overall survival in CRC. This needs to be taken into account when interpreting the results of the present study.

## 4. Materials and Methods

### 4.1. Patient Samples

The present study was performed in accordance with the Declaration of Helsinki. All protocols were approved by the Institutional Review Board at the University Hospital Erlangen, Germany (Nr. 339_15 Bc). Written informed consent was obtained from each of the participating patients prior to surgery. Patients aged 18 years or older with a postoperative pathological diagnosis of colorectal adenocarcinoma who underwent elective surgery at the Department of Surgery of the University Hospital Erlangen between 2016 and 2019 were eligible to be enrolled into the present study. Paired tumor and adjacent native colon tissues were collected from a total of 62 CRC patients. The clinicopathological characteristics of the study cohort are shown in Table 3.

### 4.2. Sample Preparation

An overview of the workflow of the present study is demonstrated in Figure 4A. Briefly, fresh resected tumor and adjacent normal control tissue samples were washed with PBS (Cat-No. 14190169, Gibco, Waltham, MA, USA) and their size and weight were measured with the lab balance ENTRIS224-1S, Sartorius. It has a readability of 0.1 mg, a repeatability of 0.1 mg, and a linearity of 0.2 mg accordingly to the manufacturer’s data sheet. Thereafter, the samples were mechanically minced into small pieces and subjected to enzymatic digestion with 60 U/mL DNase I (Cat-No. D5319, Sigma-Aldrich, St. Louis, MO, USA), 450 U/mL collagenase I (Cat-No. C0130, Sigma-Aldrich, St. Louis, MO, USA), 60 U/mL collagenase XI (Cat-No. C7657, Sigma-Aldrich, St. Louis, MO, USA), and 60 U/mL hyaluronidase (Cat-No. H1115000, Sigma-Aldrich, St. Louis, MO, USA), at 37 °C for 1 h while shaking at 750 rpm. After digestion, the samples were filtered through a 40 µm cell strainer, while being flushed with PBS to acquire cell suspension. Total viable cell numbers were determined using Trypan blue staining. Finally, the cell suspensions were centrifuged at 350× *g* for 5 min at 4 °C, and the cell concentration was adjusted to no more than 1 million cells per 100 μL by re-suspending with an appropriate volume of PBS containing 1% FBS (Cat-No. A3160802, Gibco, Waltham, MA, USA), 0.5% BSA (Cat-No. A2153, Sigma-Aldrich, St. Louis, MO, USA), and 2 mM EDTA (Cat-No. AM9260G, Invitrogen, Waltham, MA, USA) (FACS buffer) for flow cytometric analysis. 

### 4.3. Flow Cytometry

The following antibodies were used for flow cytometric analysis: anti-CD45-BV786 (Cat-No. 563716, BD Biosciences, Franklin Lakes, NJ, USA), anti-CD20-BUV395 (Cat-No. 563781, BD Biosciences, Franklin Lakes, NJ, USA), anti-CD73-BUV510 (Cat-No. 563198, BD Biosciences, Franklin Lakes, NJ, USA), anti-IgG1-BUV510 (Cat-No. 562946, BD Biosciences, Franklin Lakes, NJ, USA), anti-CD326-FITC (Cat-No. 324204, BioLegend, San Diego, CA, USA), anti-7-AAD-PerCP-Cy5.5 (Cat-No. 559925, BD Biosciences, Franklin Lakes, NJ, USA), anti-CD38-BUV737 (Cat-No. 612824, BD Biosciences, Franklin Lakes, NJ, USA), anti-IgG-BV421 (Cat-No. 400157, BD Biosciences, Franklin Lakes, NJ, USA), anti-CD27-BV711 (Cat-No. 563167, BD Biosciences, Franklin Lakes, NJ, USA), anti-CD24-FITC (Cat-No. 560992, BD Biosciences, Franklin Lakes, NJ, USA), anti-CD20-BV650 (Cat-No. 563780, BD Biosciences, Franklin Lakes, NJ, USA), anti-IgM-PE-CF594 (Cat-No. 562539, BD Biosciences, Franklin Lakes, NJ, USA), and anti-IgD-PerCP-Cy5.5 (Cat-No. 348234, Biolegend, San Diego, CA, USA). Data were acquired on a Celesta (BD Biosciences, Franklin Lakes, NJ, USA) flow cytometer using the BD FACSDiVa™ software v8.0.1.1 and analyzed with FlowJo 10.3.0 (FlowJo LLC, Ashland, OR, USA).

The gating strategies we used to identify the cells of interest are illustrated in Figure 4B. The cell count of the 7-AAD negative population on FACSCelesta was considered equivalent to the total viable cell number manually counted under microscope for the corresponding samples. The cell numbers of the daughter populations were calculated by multiplying the cell percentage in the 7-AAD negative population obtained from FlowJo software and the total viable cell number. All the cell numbers were normalized as cell number per gram tissue.

### 4.4. Histology

The resected tumor and adjacent normal control tissue samples were fixed, paraffin embedded, and sectioned as per standard protocol by the expert pathologist (C.G.) at the Department of Pathology of University Hospital Erlangen. Serial sections 4 μm in thickness were prepared with Leica Microtome and mounted on the slides. The slides were stained with hematoxylin and eosin (H&E) for overall histological evaluation according to standard protocol. For immunohistochemistry (IHC) staining, the deparaffinized and hydrated slides were subjected to antigen retrieval to expose the antigenic sites; and then were subjected to 0.3% hydrogen peroxide (H_2_O_2_) in PBS for 5 min to inactivate endogenous peroxidase and were blocked in 10% normal goat serum with 2% BSA in PBS for 30 min; followed by overnight incubation with rabbit monoclonal anti-human CD73 antibody (Abcam) diluted at 1:100 in TBS with 1% BSA. The slides were subsequently incubated with EnVision+ System horseradish peroxidase (HRP)-conjugated anti-rabbit IgG secondary antibody (Dako) for 30 min; and developed with 3,3′-Diaminobenzidine (DAB) for 12 min. The slides were then counterstained with hematoxylin to identify the nuclei, dehydrated, mounted, and coverslipped with an aqueous-based mounting medium. The images were captured by the digital slide scanner Panoramic 250 Flash II (3D Histech) at 20× magnification.

### 4.5. Statistical Analysis

Statistics and significant outliers were determined using GraphPad Prism 9.2 software. For comparison of measurement data between 2 groups, unpaired *t* test was applied. For comparison of paired flow cytometric data between tumor and adjacent normal tissues, the paired *t* test was adopted. For comparison among 3 groups, one-way ANOVA was used. For counting data, the Pearson Chi-square test was applied. Survival data was analyzed using the Log-rank (Mantel-Cox) test. Differences were considered statistically significant at *p* < 0.05. All *p* values are two-tailed.

## 5. Conclusions

The present study is the first to investigate CD73 expression on B-cells in CRC and its association with clinicopathological characteristics, suggesting that high CD20^+^CD73^+^ B-cell infiltration into the tumor tissue is correlated with a better patient outcome. As new promising immune therapies harnessing B-cell function are the focus of current research, the effects of B-cells and their subsets within the TME on the development of colorectal cancer should be further explored to find new therapeutic options.

## Figures and Tables

**Figure 1 ijms-23-05163-f001:**
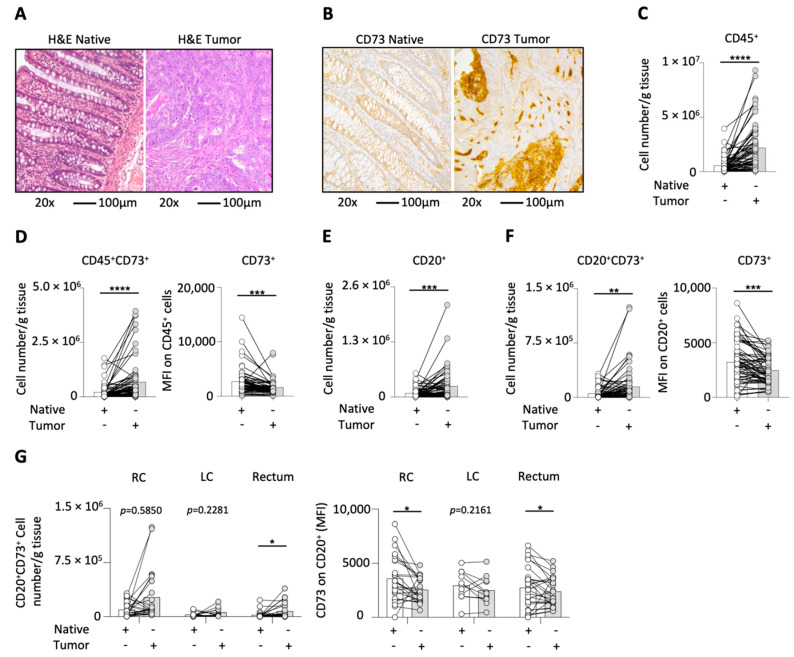
H&E (**A**) and IHC staining with CD73 (**B**) of native and tumor tissues in CRC patients; images are shown at 20× magnification; bars indicate 100 µm; CD45^+^ cell numbers were increased in tumors (**C**), CD45^+^CD73^+^ leucocytes numbers and CD73 MFI on CD45^+^ cells (**D**), numbers of infiltrated CD20^+^ B-cells (**E**), cell count of CD20^+^CD73^+^ B-cells and CD73 expression on B-cells (**F**), CD20^+^CD73^+^ B-cell numbers and CD73 MFI on CD20^+^ B-cells in CRC tumors from different locations (**G**); **** = *p* < 0.0001; *** = *p* < 0.001; ** = *p* < 0.01; * = *p* < 0.05.

**Figure 2 ijms-23-05163-f002:**
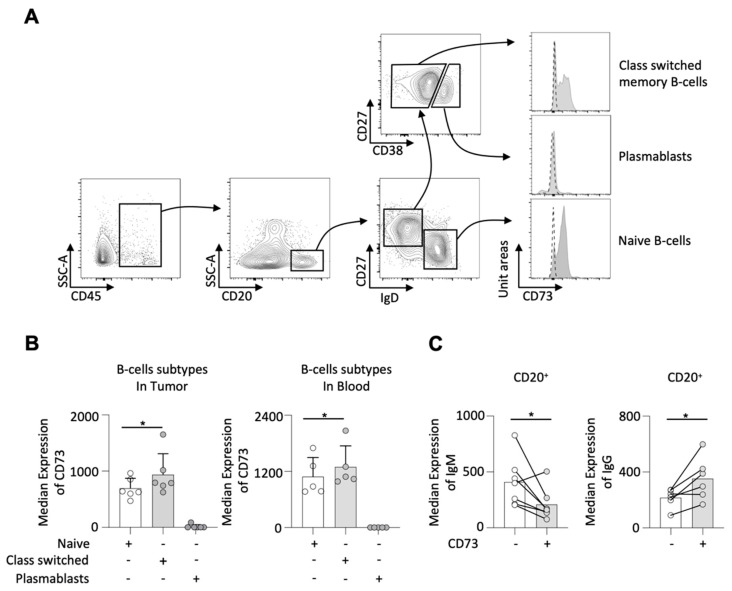
Gating strategy for identifying the different B-cells subsets and CD73 expression. Naive B-cells were defined as CD20^+^IgD^+^CD27^−^, plasmablasts as CD20^+^IgD^−^CD27^+^CD38^+^ and class-switched-memory B-cells as CD20^+^IgD^−^CD27^+^; histogram plots show the expression of CD73 on different B-cells subsets in tumor; the dotted histogram plots represent the isotype controls (**A**), the median expression of CD73 on different B-cell subsets in tumor and blood (**B**), CD73^+^ B-cells have a different Ig depot than CD73^−^ B-cells in tumor (**C**); * = *p* < 0.05.

**Figure 3 ijms-23-05163-f003:**
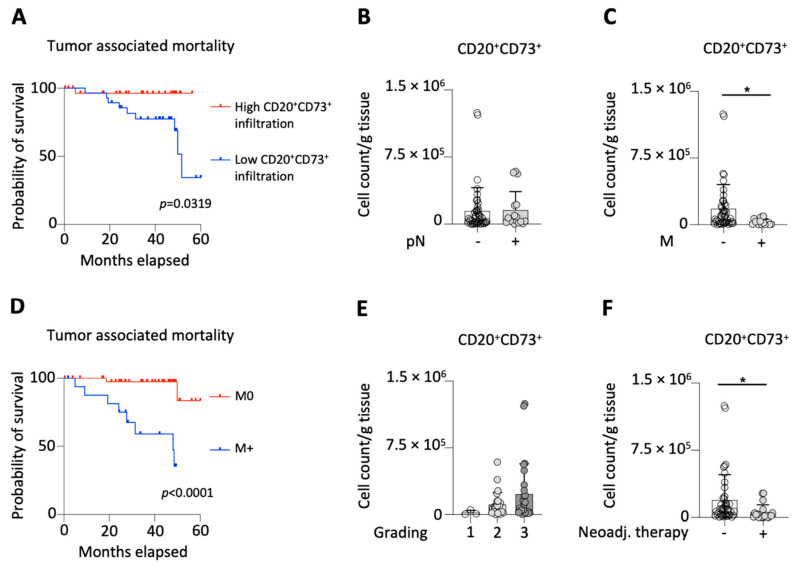
CD20^+^CD73^+^ B-cell numbers from 62 patients were correlated with the following clinicopathological parameters including: survival of patients (**A**), lymph node status (**B**), and the presence of distant metastasis (**C**); survival of localized was compared to metastasized tumors (**D**); cell count of CD20^+^CD73^+^ B-cells were also correlated with the histological grading (**E**), and the status of neoadjuvant treatment (**F**); * = *p* < 0.05.

**Figure 4 ijms-23-05163-f004:**
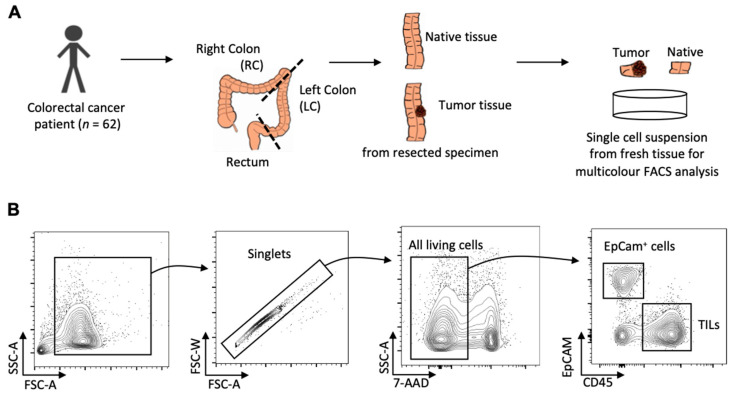
A flowchart summarizing the workflow of the study (**A**); gating strategy for identifying EpCam^+^ cells and tumor-infiltrating leucocytes (TILs) (**B**).

**Table 1 ijms-23-05163-t001:** Clinicopathological characteristics of the patients grouped according the median cell count of CD20^+^CD73^+^ B-cells into low (*n* = 31) and high (*n* = 31) infiltration.

Infiltration of CD20^+^CD73^+^		Low	High	*p*-Value
Number		31	31	
Mean Age (in years (range))		66 (30–90)	70 (38–82)	0.2713
Sex	Female	11	12	0.79
	Male	20	19	
Localization of the tumor	Right Colon	5	19	**<0.01**
	Left Colon	7	4	
	Rectum	19	8	
Mean Tumor size (in cm (range))		3.7 (0–7.2)	4.4 (0–14.0)	0.3839
Grading	G1	2	1	0.08
	G2	9	16	
	G3	13	13	
	Unknown	7	1	
pN-category	pN0	22	23	0.78
	pN+	9	8	
Distant Metastasis	No	19	26	**0.05**
	Yes	12	5	
UICC stage	I	4	10	0.12
	II	10	12	
	III	5	4	
	IV	12	5	
Neoadjuvant treatment	Yes	16	4	**0.041**
	No	15	27	
CEA	<5 µg/L	16	11	0.41
	≥5 µg/L	5	8	
	Unknown	10	12	

**Table 2 ijms-23-05163-t002:** Clinicopathological characteristics of the patients grouped according to the status of neoadjuvant therapy.

Neoadjuvant Treatment		−	+	*p*-Value
Number		42	20	
Mean cell count of CD20^+^CD73^+^ (in per gram tissue)		191,753	55,206	**0.0411**
Mean MFI of CD73 on CD20^+^		2707	2008	**0.0363**
Mean Age (in years (range))		68 (30–87)	67 (52–90)	0.7117
Sex	Female	15	8	0.74
	Male	27	12	
Localization of the tumor	Right Colon	23	1	**<0.01**
	Left Colon	11	0	
	Rectum	8	19	
Mean Tumor size (in cm (range))		4.1 (0.15–11.3)	3.6 (0–7.0)	0.4052
Grading	G1	3	0	**<0.01**
	G2	17	8	
	G3	22	4	
	Unknown	0	8	
pN-category	pN0	29	16	0.37
	pN+	13	4	
Distant Metastasis	No	35	10	**<0.01**
	Yes	7	10	
UICC stage	I	9	5	**<0.01**
	II	17	5	
	III	9	0	
	IV	7	10	
CEA	<5 µg/L	16	11	0.40
	≥5 µg/L	9	4	
	Unknown	17	5	

**Table 3 ijms-23-05163-t003:** Characteristic features of the study cohort.

	CRC Patients
Number	62
Age (median)	71 (38–90)
Sex (Male:Female)	39:23
Neoadjuvant therapy	
Radiochemotherapy	19
Chemotherapy	1
-	42
Localization of the tumor	
Right colon (RC)	24
Left colon (LC)	11
Rectum	27
Histological Grade	
G1	3
G2	25
G3	26
Unknown	8
pN-category	
pN0	17
pN+	45
Distant Metastasis	
No	45
Yes	17
UICC stage	
I	14
II	22
III	9
IV	17
Preoperative CEA level	
Low (<5 µg/L)	27
High (≥5 µg/L)	13
Unknown	22

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
