# Peer review of "Tumor Infiltration with CD20+CD73+ B Cells Correlates with Better Outcome in Colorectal Cancer"

_ijms, 2022, doi:10.3390/ijms23095163_

Round 1
Reviewer 1 Report
In the present study, the authors investigated the expression of CD73 on tumor-infiltrating B-cells in colorectal cancer (CRC) and the relationship between the B cells population expressing CD73 and prognosis of CRC patients. It is demonstrated that the class-switched memory B-cells showed highest expression of CD73 on tumor-infiltrating B-cells. In addition, it is demonstrated that the higher CD73+ B-cells infiltration in the CRC tumors was associated with better overall survival. I think that the topic and obtained results of this study is interesting. In my opinion, this manuscript is suitable for the publication in International Journal of Molecular Sciences after minor revision.
Comments:
- The information about the sources of some reagents such as DNase I is missing.
- Table 1: “umber” → “Number”
- The catalog number of antibodies used in this study should be described.
- Figure 1B and 3A: I think that the shown gating is not correct. For example, in Figure 1B, although the threshold level at X axis of FSC-A in 1st panel is about 0.8 major thick, the 2nd panel shows that the gated population contains the cells with lesser than 0.8 major scale. This means that the gating is not performed in the indicated workflow. In addition to FSC-A, the gating of SSC-A is not performed when comparing 1st panel with 3rd Similarly, I think that the gating was not performed in Figure 3 because there are inconsistent in SSC-A level.
- Figure 2C-G: How did the authors obtain the cell number of each population? In addition, it is unknow how the authors measured the weight of tissues, and its accuracy.
- Figure 3A: What is the dotted line? Isotype control?
- In the present study, the authors selected MFI for the estimation of CD73 intensity. However, since Figure 3A showed that histogram is not normally distributed, I think that median is suitable for the estimation of CD73 intensity.
Reviewer 2 Report
The authors provide tumor infiltration B-cell analysis in colorectal cancer. They correlated the subset of B-cells with patient survival. The work is essential in understanding the tumor microenvironment, and CD73 could potentially be a targetable molecule.
Was the author able to see CD20/CD73 cells in the histological sections? Figure 2 shows cd73 staining, which appears to be in the tumor cells. Does cd73 have different functions in the tumor cells?
The expression of CD73 is lower in CD20 cells. What implication does that have for the patients? Do patients have a better chance for survival with higher expression in B cells? How much CD73 expression is in the EPCAM positive cells? Does that have any bearing on the survival of patients in the study since it is associated with immune escape?.
There is a reduction in the CD20/CD73 B-cells with Chemotherapy. Can the author clarify whether this was associated with a worse chance for overall survival?
The small typo in table 1 “umber” should be “number”.
